# Deep-Learning-Based Visualization and Volumetric Analysis of Fluid Regions in Optical Coherence Tomography Scans

**DOI:** 10.3390/diagnostics13162659

**Published:** 2023-08-12

**Authors:** Harishwar Reddy Kasireddy, Udaykanth Reddy Kallam, Sowmitri Karthikeya Siddhartha Mantrala, Hemanth Kongara, Anshul Shivhare, Jayesh Saita, Sharanya Vijay, Raghu Prasad, Rajiv Raman, Chandra Sekhar Seelamantula

**Affiliations:** 1Department of Electrical Engineering, Indian Institute of Science, Bengaluru 560012, India; harishwarreddy98@gmail.com (H.R.K.); udaykanthreddy111@gmail.com (U.R.K.); siddhartha.mantrala@gmail.com (S.K.S.M.); hemanthk@alum.iisc.ac.in (H.K.); anshulshivha@iisc.ac.in (A.S.); 2Carl Zeiss India Pvt. Ltd., Bengaluru 560099, India; jayesh.saita@zeiss.com (J.S.); sharanya.vijay.ext@zeiss.com (S.V.); raghu.prasad@zeiss.com (R.P.); 3Shri Bhagwan Mahavir Vitreoretinal Services, Sankara Nethralaya, Chennai 600006, India; rajivpgraman@gmail.com

**Keywords:** optical coherence tomography, deep learning, visualization, classification, fluid volume computation

## Abstract

Retinal volume computation is one of the critical steps in grading pathologies and evaluating the response to a treatment. We propose a deep-learning-based visualization tool to calculate the fluid volume in retinal optical coherence tomography (OCT) images. The pathologies under consideration are Intraretinal Fluid (IRF), Subretinal Fluid (SRF), and Pigmented Epithelial Detachment (PED). We develop a binary classification model for each of these pathologies using the Inception-ResNet-v2 and the small Inception-ResNet-v2 models. For visualization, we use several standard Class Activation Mapping (CAM) techniques, namely Grad-CAM, Grad-CAM++, Score-CAM, Ablation-CAM, and Self-Matching CAM, to visualize the pathology-specific regions in the image and develop a novel Ensemble-CAM visualization technique for robust visualization of OCT images. In addition, we demonstrate a Graphical User Interface that takes the visualization heat maps as the input and calculates the fluid volume in the OCT C-scans. The volume is computed using both the region-growing algorithm and selective thresholding technique and compared with the ground-truth volume based on expert annotation. We compare the results obtained using the standard Inception-ResNet-v2 model with a small Inception-ResNet-v2 model, which has half the number of trainable parameters compared with the original model. This study shows the relevance and usefulness of deep-learning-based visualization techniques for reliable volumetric analysis.

## 1. Introduction

Intraretinal Fluid (IRF), Subretinal Fluid (SRF), and Pigmented Epithelial Detachment (PED) are three of the most prevalent retinal pathologies [1,2,3]. Quantitative measures of retinal fluids are important biomarkers, and fluid volume computation from optical coherence tomography (OCT) scans by artificial intelligence (AI) algorithms can guide the treatment of exudative retinal conditions, such as Neovascular Age-related Macular Degeneration (NAMD). In NAMD, pathological fluids accumulate in different compartments of the retina—IRF inside the retina, SRF beneath the neurosensory retina, and PED between the Retinal Pigment Epithelium (RPE) and Bruch’s membrane (BM). Different types of OCT technologies are used to capture the retina of the eye, such as time-domain, spectral-domain, and swept-source OCT [4]. In this paper, we use data generated from commercially available CIRRUS and PRIMUS OCT machines, which are spectral-domain-based OCT machines.

OCT captures structural information of the retina using the principle of coherent detection. Coherent detection produces multiplicative noise, known as speckle in the acquired images. Speckle lowers image quality and makes visual and automated analysis of OCT images challenging [5]. The noise statistics in OCT images play a significant role in developing suitable despeckling algorithms. Various data distributions have been proposed in the literature for modeling speckles in OCT images. According to Bashkansky and Reintjes [6], speckle noise follows a Gaussian distribution. Pircher et al. [7] and Karamata et al. [8] suggested that speckle noise follows a Rayleigh distribution. Schmitt et al. [9] proposed an exponentially decaying distribution for speckle. Sudeep et al. [10] proposed a denoising method based on the Gamma distribution for speckle reduction in OCT images. In this paper, we model the speckle noise as gamma distribution and perform denoising of OCT images as the first step.

Deep learning has produced remarkable advancements in the field of healthcare [11,12,13,14], especially in the automatic detection of retinal abnormalities such as glaucoma, diabetic retinopathy, Diabetic Macular Edema (DME), Age-related Macular Degeneration (AMD), and Retinal Vein Occlusion (RVO). Deep-learning-based classifiers are used to detect the relationship between the multiple input features in the image and analyze the relationships between these features to make predictions or classifications. By leveraging the complex patterns and representations learned by the model, it can provide an overall prediction based on these detected features. However, for deploying these classifiers in the real world, there is a need to provide an explanation of the relationships captured among the features of the image. This problem of explainable artificial intelligence (XAI) can be addressed to a certain extent by using appropriate visualization tools. Techniques such as Grad-CAM [15], Grad-CAM++ [16], Score-CAM [17], Ablation-CAM [18], and Selfmatching- CAM [19] are used to explain which features of the image are significant contributors to the prediction score. Explainability enhances the confidence of doctors in deploying AI models in real-world applications. The models are otherwise considered opaque or black-box type. Explainability in AI has now become an indispensable part of the deployment pipeline, because when deploying a deep learning model in real-world scenarios, it is crucial to be able to explain and interpret the model’s decisions or predictions, particularly when dealing with medical applications. Explainability in AI is necessary to provide insights and justifications to doctors or medical professionals, enabling them to understand why a certain deep learning model is classifying an image into a specific pathology. Through explainability, doctors can gain trust and confidence in the model’s capabilities, facilitating its adoption and integration into clinical practice.

There are several studies available in the literature that proposed automatic volume computation of retinal fluid. Typically, one performs segmentation for calculating the fluid volume, for instance, Yukun et al.’s Retinal Fluid Segmentation Network (ReF-Net) [20], Thomas et al.’s encoder–decoder-based segmentation network [21], Wilson et al.’s U-Net-based architecture [22], and Lu et al.’s automated segmentation technique [23]. To the best of our knowledge, ours is the first attempt at adopting the visualization outcome of a deep learning model for retinal fluid volume computation instead of a segmentation output. Typically, one employs a segmentation-based approach for calculating the retinal fluid volume. However, our methodology is significantly different, as we utilize the results obtained from visualization techniques. We also present a Graphical User Interface (GUI) for automatic fluid volume computation. We consider GradCAM, GradCAM++, Score-CAM, Ablation-CAM, and Self-Matching-CAM as visualization techniques for the binary classification models, which detect IRF, SRF, and PED pathologies in the retinal OCT images. The volume is computed separately for IRF, SRF, and PED. In addition, we also propose a novel fusion technique, namely Ensemble-CAM, which is a robust visualization technique.

A combination of IRF, SRF, and PED, as well as fluid volume variations, are used to categorize Age-related Macular Degeneration (AMD) patient response to antivascular endothelial growth factor (Anti-VEGF) therapy. The AMD patient is categorized as a Responder if the post-treatment fluid volume has reduced by more than 10% compared with the pretreatment fluid volume and as a Nonresponder if the post-treatment fluid volume has increased or remained the same or reduced by less than 10% compared with the pretreatment fluid volume. The automatic quantification of fluid response may be a better means of monitoring than the current qualitative evaluation used in clinical practice. The computed fluid volume could be a potential input to an AI model to predict the AMD patient’s response to an Anti-VEGF injection [24]. The AI response prediction can help clinicians in deciding the form of the treatment and could be a beneficial tool in the clinical armamentarium for the therapeutic management of AMD.

Section 2 contains a description of the dataset used for classification and visualization tasks and the methodology used in the paper. Section 3 presents the results obtained for classification, visualization, and fluid volume computation, and Section 4 contains a discussion of the results. Concluding remarks are given in Section 5.

Figure 1 shows an OCT image in which all three pathologies, namely IRF, SRF, and PED, are present.

## 2. Materials and Methods

### 2.1. Dataset

In this section, we present details of the datasets used for classification and visualization.

#### 2.1.1. Classification Dataset

The dataset used in this study is proprietary to Carl Zeiss and is the same as the one used in our recent publication [25]. It comprises OCT B-scan images stacked as cubes. The images are obtained using two brands of Carl Zeiss machines—CIRRUS and PRIMUS. There are, in turn, two variants of CIRRUS, one that captures 200 images per cube and an other that captures 128 images per cube. There are two types of PRIMUS machines, of which one captures 128 images per cube and an other that captures 32 images per cube. The training and test data for classification consist of 207,535 and 52,650 images, respectively. The B-scan images from CIRRUS and PRIMUS are of size 1024 × 512 and 1024 × 200, respectively.

#### 2.1.2. Segmentation Dataset

The ground-truth segmentation data required for visualization is available for 73 cubes, each, in turn, containing 128 B-scans, making up a total of 9344 (73 × 128) images. Of these, 1235, 1361, and 3686 images have IRF, SRF, and PED pathologies, respectively.

#### 2.1.3. Performance Measures

The segmentation ground truth is given by an expert and is used for the computation of the Jaccard Index/Intersection over Union (IoU) [26] and Dice score [27,28] given as follows:(1)$$Jaccard\left(A,B\right)=\frac{\left|A\cap B\right|}{\left|A\cup B\right|},$$
and
(2)$$Dice\left(A,B\right)=\frac{2|A\cap B|}{\left|A|+|B\right|},$$
where *A* is the generated mask, *B* is the ground truth mask, and |·| represents the cardinality of the set.

### 2.2. Architecture of the Model

In this work, we use the Inception-ResNet-v2 [29] to develop the binary classification models for each pathology. Figure 2a shows the standard Inception-ResNet-v2 architecture divided into several parts, such as stem, Inception-A, Inception-B, Inception-C, Reduction-A, and Reduction-B, each explained in the following:**Stem**: The stem of the Inception-Resnet-v2 architecture consists of a convolutional layer followed by a max pooling layer, and several convolutional layers with increasing dilation rates.**Inception-ResNet-A**: The Inception-A blocks consist of multiple branches, each of which applies a different type of operation (such as convolution or pooling) to the input. These branches are then concatenated together and passed through a final convolutional layer.**Inception-ResNet-B**: The Inception-B blocks are similar to Inception-A blocks, but with a different set of operations. It uses an average pooling operation followed by a convolutional layer.**Inception-ResNet-C**: Inception-C blocks are also similar to Inception-B blocks but with a different set of operations and a larger number of filters.**Reduction**: Reduction-A and Reduction-B blocks reduce the spatial dimensions of the feature maps. They consist of several convolutional layers and max pooling layers.

Finally, the output of the last block is fed into an average pooling layer, followed by a dropout layer and a fully connected layer, which produces the final output. The Inception-ResNet-v2 is trained on 1 million images from ImageNet [30] dataset with 1000 classes. We fine-tuned the model using the training dataset with two classes.

### 2.3. Comparison with Small Inception-ResNet-v2

We developed a novel small Inception-ResNet-v2 architecture, which has a similar skeleton to the original Inception-ResNet-v2 but with half the number of trainable parameters (28 million parameters as opposed to 56 million trainable parameters). Ten Inception-ResNet-A, twenty Inception-ResNet-B, and ten Inception-ResNet-C layers in Inception-ResNet-v2 architecture are replaced by eight Inception-ResNet-A, seven Inception-ResNet-B, and four Inception-ResNet-C layers, respectively, to obtain the small Inception-ResNet-v2 architecture shown in Figure 2b. Figure 2c shows the stem.

### 2.4. Denoising

The OCT images obtained from the OCT scanners are typically noisy. Hence, there is a need to perform mild denoising of the images. If the models are trained directly on noisy images, there is a chance that the model might fit the noise in the image rather than the pathological features of interest. After experimenting with several noise models, we found the gamma distribution to be an appropriate model for the noise in OCT images. We carry out denoising using the Gated Convolution and Deconvolution Structure (GCDS) model [5]. Figure 3a,b shows the noisy OCT image and the denoised OCT image, respectively, obtained from the GCDS model. The advantages of the GCDS model are as follows:The GCDS model consists of two phases—an encoding phase and a decoding phase. The encoding phase consists of 5 convolution layers that create a representation that encapsulates all fundamental features but leaves out the noise.The model also contains skip connections between corresponding convolution and deconvolution layers. These skip connections reduce the number of weights to be trained in the neural network, which leads to quicker convergence of the model.Each skip connection is associated with a gating factor, which determines the ratio of split of the information between the next convolution layer and the corresponding deconvolution layer.

### 2.5. Classification of the OCT Images

We now present the details involved in training the classification model.

Figure 4 shows the pipeline for training a classification model. From here on, we refer to the model trained on noisy images as the noisy model; the model trained on denoised images obtained from the GCDS model using original Inception-ResNet-v2 architecture as the Inception-ResNet-v2 GCDS model; and the model trained on denoised images obtained from the GCDS model using small Inception-ResNet-v2 architecture as the small Inception-ResNet-v2 GCDS model. For the Inception-ResNet-v2 GCDS model and small Inception-ResNet-v2 GCDS model training, noisy OCT images are denoised and used for training the classification model. Post training, we evaluate the performance of the model on the test dataset and compute the classification metrics. For training the noisy model, the denoising step is bypassed, and all the other steps remain the same.

#### Training Phase

We trained the classification model on 207,535 images. Each image carries labels for IRF, SRF, and PED. For training the classifier, we resize the images, which are of size (1024, 512, 3) to (450, 450, 3) for compatibility with the Inception-ResNet-v2 architecture. The model is tested using data consisting of 52,650 images. We used an Inception-ResNet-v2 architecture pretrained on the ImageNet dataset. We fine-tuned the pretrained model using OCT training images. We used the Adam optimization algorithm [31], which combines adaptive gradient methods and stochastic optimization techniques. Adam has been shown to be effective in various deep learning tasks, as it dynamically adjusts the learning rate based on the gradient magnitudes of individual parameters.

We set the learning rate to $10^{-4}$, which was chosen based on empirical observations and previous research in the field. This learning rate provides a good balance between convergence speed and stability, ensuring that the model converges to a suitable solution while avoiding overshooting or instability issues.

The training phase comprises iterative updating of the model parameters using backpropagation and stochastic gradient descent. The Adam optimizer adjusted the learning rate for each parameter separately, allowing the model to effectively adapt to different features and gradients present in the training data. The loss function employed is Categorical Cross-Entropy (CCE) [32] given by
(3)$$CCE\;Loss=-\sum_{i=1}^{output\;size}y_{i}\cdot \log{\hat{y}}_{i},$$
where ${\hat{y}}_{i}$ is the *i*th entry (scalar value) in the model output, $y_{i}$ is the corresponding target value, and the output size is the number of scalar values in the model output.

### 2.6. Visualization of the OCT Images

Due to the black-box nature of the deep-learning-based classification models, there is a need for explainability of the model. We consider standard visualization-based techniques such as GradCAM, GradCAM++, Score-CAM, Ablation-CAM, and Self-Matching-CAM on the binary classification models of IRF, SRF, and PED. These visualization techniques help doctors understand why the model made a certain decision for an image by highlighting relevant parts of the image. Figure 5b shows the Grad-CAM output of the GCDS denoised OCT image shown in Figure 5a. In the heat map, the yellow region indicates the most relevant region, and the relevance decreases as the color changes from yellow to blue.

#### Ensemble CAM

We applied standard visualization techniques such as Grad-CAM (GD), Grad-CAM++ (GD++), Score-CAM (SC), Ablation-CAM (AC), and Self-Matching-CAM (SM) on the nine binary classification models developed for each of the three pathologies using the noisy OCT images and denoised OCT images. The heat maps obtained from the visualization techniques are converted to binary maps using the Otsu thresholding [33] technique. The procedure is applied to expert-annotated ground-truth segmentation comprising 128 B-scans, making up a total of 9344 (73 × 128) images to calculate the Intersection over Union (IoU) and Dice scores.

We observed that for several OCT images, the heat maps given by various visualization techniques are not always in agreement, as the techniques emphasize different regions in the OCT image. The Ensemble CAM output is a binary map with the value at a certain pixel equal to one, where three or more than three visualization techniques agree with each other. Computationally, the Ensemble CAM heat map can be obtained by simply adding the five binarized class activation maps and using binary thresholding with a threshold value of three. Figure 6 shows the binary maps obtained from various visualization techniques, including the Ensemble-CAM that we developed.

### 2.7. Retinal Fluid Volume Computation

AI-based fluid volume computation compared with manual-annotations-based volume computation saves an enormous amount of time and removes intraobserver and interobserver variability. Currently, the device does not provide clinicians the ability to measure fluid volumes automatically, and the assessment is performed by visual inspection. An automatic fluid volume computation tool would greatly enable the clinicians in providing reliable diagnosis.

The fluid volume is computed using two different techniques, namely the region-growing algorithm [34,35] and the thresholding technique. We computed the volumes of 73 C-scans for IRF, SRF, and PED pathologies separately using the region-growing and thresholding techniques.

Figure 7 shows the pipeline for computing the volume of IRF, SRF, and PED in a C-scan. First, forthe Inception-ResNet-v2 GCDS model and the small Inception-ResNet-v2 GCDS model, we pass the noisy OCT images through the denoising model and preprocess them before passing them to the classification model. If the prediction score is more than 0.5, we pass the preprocessed image through the visualization technique to obtain the heat map, which is further binarized using the Otsu thresholding technique [33]. Figure 8 shows the snapshots of the Graphical User Interface (GUI) created for volume computation. The doctors select a C-scan and press the volume computation button to display the results. Four images are displayed in the GUI—the input image, the CAM output, the binary map obtained from the CAM heat map, and the ground truth corresponding to the input image.

### 2.8. Post Processing

We used a classification model to gain insights into the explainability of the model’s predictions. To visualize the important regions contributing to the model’s decision-making process, we employed several popular techniques, such as GradCAM, GradCAM++, ScoreCAM, LayerCAM, and AblationCAM. These techniques generate heat maps that highlight the regions of interest in the input image. However, we observed that these heat maps only provided an indication of the regions without precisely outlining the boundaries of the specific regions of interest. This limitation became apparent when dealing with conditions such as IRF (Intraretinal Fluid) and SRF (Subretinal Fluid), where it is crucial to localize the presence of fluid in the retina precisely.

We explored two additional techniques to address this issue, selective thresholding and region-growing, with the aim of refining the heat maps generated by the CAM techniques by emphasizing the regions corresponding to fluid-containing areas. We noticed that these regions appeared darker in contrast to the other areas. By applying selective thresholding and region-growing techniques, we improved the localization of the fluid-contained regions and obtained more accurate outlines of these regions. These techniques allowed us to enhance the interpretability of the classification model for identifying and precisely delineating the presence of fluid in the retina, specifically for IRF and SRF.

#### 2.8.1. Region-Growing Technique

To further enhance the identification and delineation of regions of interest, we employed the region-growing technique [36,37], leveraging the CAM output from the classification model. The purpose was to identify potential regions within the image that corresponded to specific features, particularly in the case of Intraretinal Fluid (IRF) and Subretinal Fluid (SRF).

The region-growing approach comprises the following steps:**Thresholding and contour extraction**: The CAM output was thresholded to convert it into a binary image. Contours were then extracted from the binary image. The centroids of these contours were considered as seed points, representing potential regions of interest within the original input image.**Seed point selection**: The centroids of the extracted contours served as the seed points for region expansion. These seed points acted as starting points for the iterative growth of the regions.**Region-growing algorithm**: Starting from the seed points, a region-growing algorithm was applied to expand the regions of interest iteratively. This algorithm examined the neighboring pixels of each seed point and determined their inclusion in the growing region based on predefined criteria.**Inclusion criteria**: The decision to include neighboring pixels in the growing region was based on a specific criterion, which involved evaluating the pixel intensity difference between the seed pixel and the corresponding neighboring pixel. If the difference was below a predefined threshold, such as 15, the neighboring pixel was considered similar and included in the region. Otherwise, it was excluded.

By iteratively growing the regions based on the inclusion criteria, we could accurately outline the desired regions of interest, specifically the fluid-contained areas in the case of IRF and SRF. This enabled precise visualization and analysis of the extent and location of these regions within the retinal images.

It is important to note that while the region-growing technique demonstrated effectiveness in identifying and delineating fluid regions, its applicability may vary for different conditions. For example, in the case of Pigment Epithelial Detachment (PED), where the objective is to highlight layer detachment areas rather than fluid regions, alternative techniques may be necessary to represent the specific features of interest accurately.

In summary, the region-growing technique, utilizing the CAM output and contour extraction, allowed us to iteratively expand regions of interest and achieve accurate outlines of fluid-contained areas. However, its suitability should be carefully considered in the context of the specific condition under analysis. Alternative techniques may be required to address different types of features and meet the specific requirements of the analysis.

#### 2.8.2. Selective Thresholding

In order to enhance the visualization and outline of regions of interest, we employed a technique known as selective thresholding. This approach aimed to improve upon the existing CAM-based visualization methods, such as GradCAM, GradCAM++, ScoreCAM, LayerCAM, and AblationCAM, which provided highlight regions but lacked precise outlines. The specific goal was to highlight fluid-containing regions in the case of IRF and SRF, while recognizing that a different approach was necessary for PED.

The selective thresholding process involved the following steps:**Obtaining the CAM output**: We obtained the CAM output from the classification model, which provided an indication of the regions of interest within the image.**Applying Otsu thresholding**: To convert the CAM output into a binary image, we applied the Otsu thresholding technique. This process assigns a value of 1 to pixels that exceed a certain threshold and 0 to pixels that are below the threshold. The result is a binary mask that highlights potential regions of interest.**Pixel-wise product**: We then applied a pixel-wise product operation between the original image and the binary mask. This operation allowed us to select and isolate the regions in the image where the mask pixel values were 1. By doing so, we focused only on the areas indicated by the CAM as regions of interest.**Histogram analysis**: To further refine the selection, we analyzed the histogram of the image. This analysis revealed that pixel values in the range of 25 to 50 exhibited a significant spike, indicating the presence of the desired regions.**Thresholding the image**: Based on the observation from the histogram analysis, we applied a thresholding operation to the image. Pixels with values in the range of 20 to 50 were considered as 1, representing the regions of interest, while the remaining pixels were set to 0. This thresholding step allowed us to achieve a more precise outline of the desired regions.**Performance evaluation**: We evaluated the effectiveness of selective thresholding by assessing metrics such as Intersection over Union (IoU) and Dice score. The results indicate a significant improvement in both metrics for the detection of fluid-containing regions in IRF and SRF cases. However, it should be noted that selective thresholding was not suitable for PED, as the objective in the case of PED was to highlight layer detachment rather than fluid regions.

In summary, selective thresholding provided a refined visualization approach by emphasizing the outline and extent of fluid-containing regions in the retinal images. While it demonstrated promising results for specific conditions, its applicability varied depending on the nature of the features of interest. For PED, alternative techniques were required to accurately highlight the layer detachment areas. Figure 9 depicts the procedure adopted for calculating the volume using the selective thresholding technique.

The volume computation process involves employing the region growing or selective thresholding techniques for all the images in a C-scan to derive masks for each True Positive image. Each pixel within the mask is considered as a cuboid, and the volume of the entire scan is computed by summing up all the individual pixel volumes given in Equation (Equation 4) across all the identified pixels in the C-scan. The volume occupied by each pixel $v_{pixel}$ [38] is scanner-specific, and for the OCT scanners under consideration, it is given by
(4)$$v_{pixel}=11.7\times 47.2\times 2.0\;\mu m^{3}.$$

## 3. Results

### 3.1. Classification Results

Metrics such as True Positive, False Positive, True Negative, False Negative, Accuracy, Precision, Sensitivity, Specificity, and F1 score are used to evaluate the classification performance. We refer to the positive labels as images in which the pathology is present and negative labels in which the pathology is absent. True Positives (TP) are the images for which prediction is abnormal, and the ground truth is also a positive label. In the case of True Negative (TN) images, the prediction is normal, and the ground truth is also a negative label. For False Positive (FP) cases, the prediction is abnormal, but the ground truth is a negative label, whereas for False Negative (FN) cases, the prediction is normal, but the ground truth is a positive label.

Accuracy is the ratio of total number of correct predictions to the total number of images and is expressed as
(5)$$Accuracy=\frac{TP+TN}{TP+FP+TN+FN}.$$

Precision is the ratio of the total number of predictions that are classified correctly as positive to the total number of positive predictions:(6)$$Precision=\frac{TP}{TP+FP}.$$

Sensitivity or Recall is the ratio of the number of images that are correctly classified as positive to the total number of positive images:(7)$$Sensitivity/Recall=\frac{TP}{TP+FN}.$$

Specificity is the ratio of the number of images that are correctly classified as negative to the total number of negative images, given by
(8)$$Specificity=\frac{TN}{TN+FP}.$$

F1 score is the harmonic mean between precision and recall and is given by
(9)$$F1\;score=\frac{2}{\frac{1}{Precision}+\frac{1}{Recall}}=\frac{2TP}{2TP+FP+FN}.$$

F1 score is mainly used when the dataset has a class imbalance.

Table 1, Table 2 and Table 3 present the classification metrics for the IRF, SRF, and PED pathologies.

### 3.2. Visualization Results

In this section, we present the quantitative visualization results of the Grad-CAM, Grad-CAM++, Score-CAM, Ablation-CAM, Self-Matching CAM, and Ensemble-CAM visualization techniques on the noisy model, Inception-ResNet-v2 GCDS model, and small Inception-ResNet-v2 GCDS model. We used IoU and Dice metrics for evaluating the visualization results. Figure 10, Figure 11 and Figure 12 show the visualization outputs obtained using the Grad-CAM technique, binary maps, and ground truths for IRF, SRF, and PED, respectively.

### 3.3. Volume Computation Results

We computed the volume over 73 C-scans obtained from the CIRRUS machine for all three kinds of models for IRF, SRF, and PED and calculated the IoU mean and standard deviation, Dice mean and standard deviation, predicted volume mean and standard deviation, and ground truth volume mean and standard deviation.

## 4. Discussion

The evaluation of the model’s performance includes various classification metrics, such as accuracy, precision, sensitivity, specificity, and F1 score. The classification results for each pathology are presented in Table 1, Table 2 and Table 3. In the case of IRF, the GCDS model of Inception-ResNet-v2 outperformed the other two models. Despite reducing the number of trainable parameters to 28 million from the original 56 million, the model’s performance metric in terms of F1 score only marginally dropped from 0.8353 to 0.8201. The noisy model’s performance lies between that of the Inception-ResNet-v2 GCDS model and the small Inception-ResNet-v2 GCDS model.

For SRF, the small Inception-ResNet-v2 GCDS model exhibited significant improvement over the original Inception-ResNet-v2 GCDS model, with the model’s performance increasing from 0.7835 to 0.8443. The noisy model’s performance metric in terms of F1 score was comparable to that of the small Inception-ResNet-v2 GCDS model. Regarding PED, the small Inception-ResNet-v2 GCDS model outperformed both the Inception-ResNet-v2 GCDS model and the noisy model.

Table 4, Table 5 and Table 6 present the visualization metrics at a B-scan level, where the Intersection over Union (IoU) and Dice coefficient are calculated between the binary map obtained from Otsu thresholding of the Grad-CAM, Grad-CAM++, Score-CAM, Ablation-CAM, Self-Matching-CAM, and Ensemble-CAM heat maps, and the segmentation data masks. For IRF, the small Inception-ResNet-v2 GCDS model achieved the best IoU using Grad-CAM++. In the case of SRF, the noisy model achieved the best IoU using the newly proposed Ensemble-CAM visualization technique. For PED, the small Inception-ResNet-v2 GCDS model obtained the best IoU using the Self-Matching-CAM visualization technique.

Table 7, Table 8, Table 9, Table 10, Table 11, Table 12, Table 13, Table 14, Table 15, Table 16, Table 17, Table 18, Table 19, Table 20, Table 21, Table 22 and Table 23 present the Intersection over Union (IoU) and Dice coefficient on a volumetric level for the region-growing and selective thresholding technique. The mean and standard deviation of these metrics are calculated for both the predicted and ground truth volume. The results are presented for all three types of models: the noisy model, the Inception-ResNet-v2 GCDS model, and the Small Inception-ResNet-v2 GCDS model. While all the models performed reasonably well on IRF and SRF, their performance in terms of volume computation for PED was below par.

**IoU and Dice results**: In certain cases of performance evaluation, such as in PED, we observed that the standard deviation is greater than the mean. This indicates that the model’s performance is inconsistent, and there is a large variability in the results obtained across different evaluations or samples. It also suggests that the model’s performance varies widely from one test instance to another, indicating outliers, instability, and unpredictability. On the other hand, for IRF and SRF, this anomaly was not observed, indicating that the model’s performance is relatively consistent, with the results clustering closer to the mean. A lower standard deviation indicates that the model’s performance is more stable and less prone to significant variations across different evaluations. In summary, when the standard deviation is greater than the mean, it implies inconsistency and variability in the performance of the model, whereas when the mean is greater than the standard deviation, it suggests stabler and consistent model performance across different evaluations. The specific interpretation may vary based on the context of the evaluation and the performance metric used.

**Fluid volume computation**: If the data points in the dataset have a wide range and are spread out, it results in a large standard deviation. This can happen when there are extreme values (outliers) that are significantly different from the rest of the data. In our case, for some patients, the fluid volume calculated was pretty high, making them outliers. Hence, the standard deviation is more than the mean in some cases.

## 5. Conclusions

In this paper, we presented a technique to compute the retinal fluid volume starting from deep-learning-based visualization modules such as Grad-CAM, Grad-CAM++, Score-CAM, Ablation-CAM, Self-Matching-CAM, and Ensemble CAM optimized on 73 C-scans obtained from CIRRUS and PRIMUS OCT machines. The visualization techniques are applied to the models trained on the Inception-ResNet-v2 and small Inception-ResNet-v2 architectures. The calculated volume is obtained using two techniques based on the region-growing algorithm and selective thresholding scheme and compared. Automatic volume computation using AI helps in reducing the effort put in by the annotators and reduces intraobserver and interobserver variability.

## Figures and Tables

**Figure 1 diagnostics-13-02659-f001:**
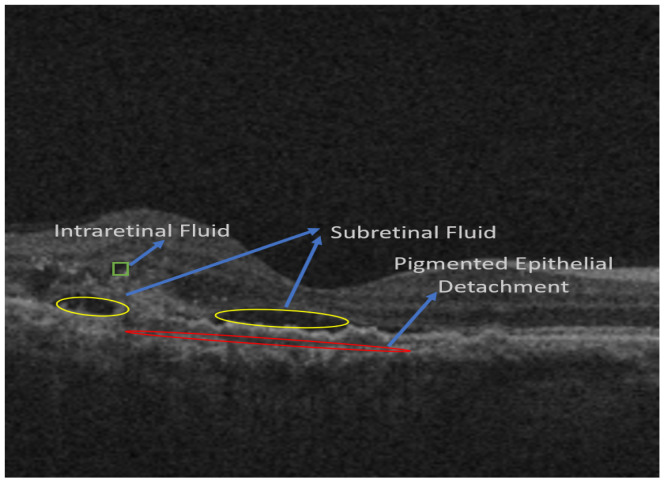
Various pathologies of interest that are observed in OCT images: IRF: Intraretinal Fluid; SRF: Subretinal Fluid; and PED: Pigmented Epithelial Detachment.

**Figure 2 diagnostics-13-02659-f002:**
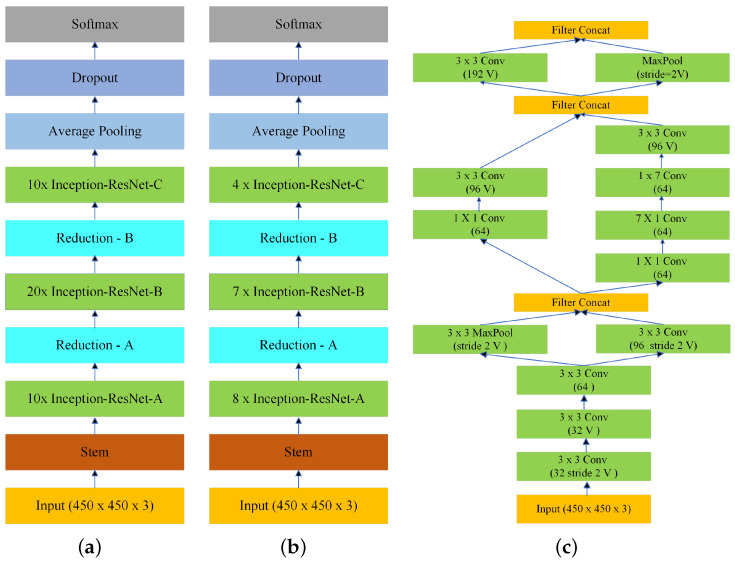
Block diagram of (**a**) Inception-ResNet-v2 architecture; (**b**) Small Inception-ResNet-v2 architecture; and (**c**) Stem.

**Figure 3 diagnostics-13-02659-f003:**
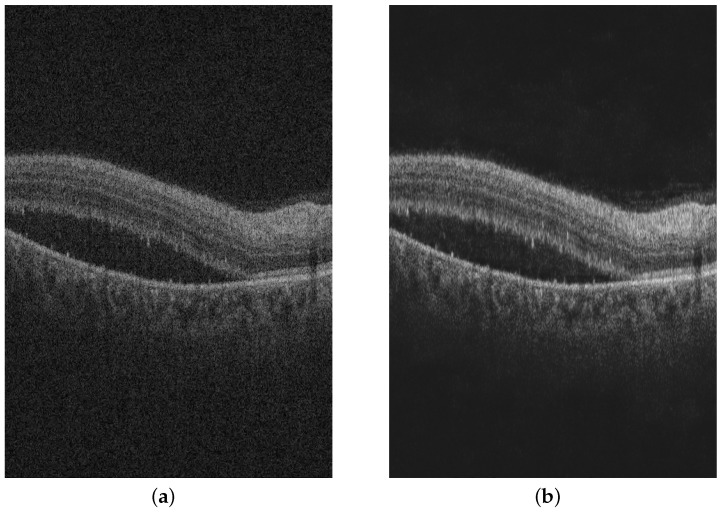
(**a**) Noisy OCT image and (**b**) Denoised OCT image using GCDS.

**Figure 4 diagnostics-13-02659-f004:**
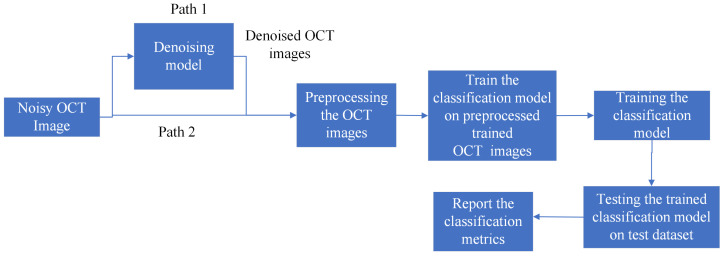
The pipeline for the training classification model: Path-1 is the pipeline for the Inception-ResNet-v2 GCDS model and small Inception-ResNet-v2 GCDS model, whereas Path-2 is the pipeline for the noisy model.

**Figure 5 diagnostics-13-02659-f005:**
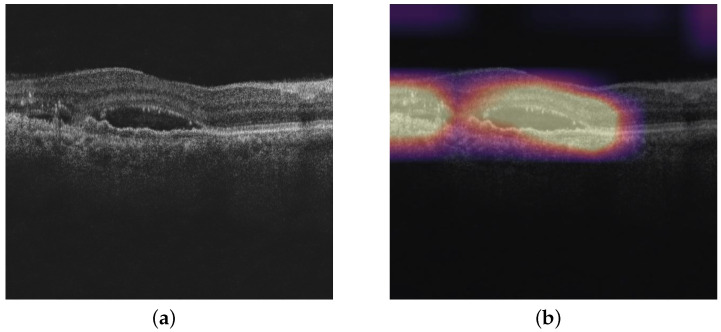
(**a**) GCDS denoised OCT image and (**b**) GradCAM output.

**Figure 6 diagnostics-13-02659-f006:**
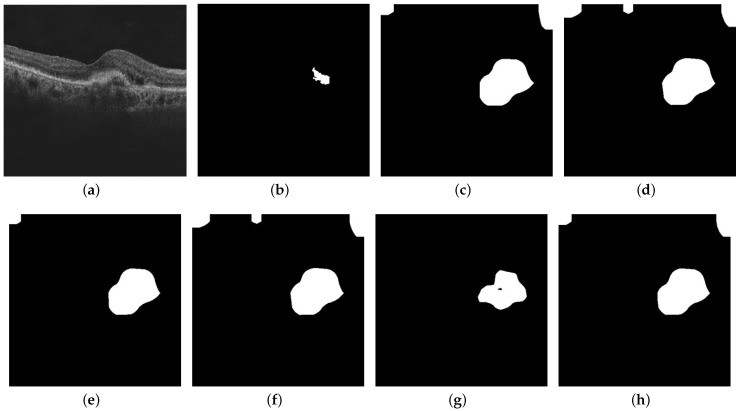
This figure shows the outputs from the Inception-ResNet-v2 GCDS model: (**a**) OCT image; (**b**) ground truth; (**c**) Grad-CAM binary map; (**d**) Grad-CAM++ binary map; (**e**) Score-CAM binary map; (**f**) Ablation-CAM binary map; (**g**) Self-Matching-CAM binary map; and (**h**) Ensemble-CAM binary map.

**Figure 7 diagnostics-13-02659-f007:**
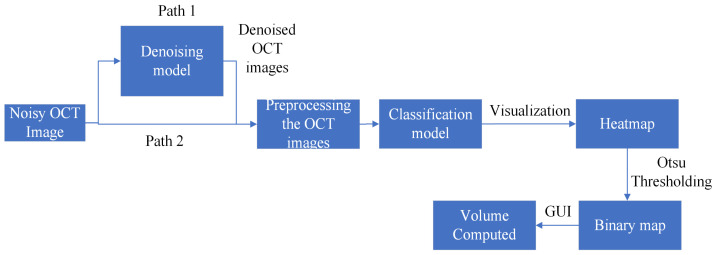
This figure shows the pipeline for volume computation: Path-1 is the pipeline for the Inception-ResNet-v2 GCDS model and small Inception-ResNet-v2 GCDS model, while Path-2 is the pipeline for the noisy model.

**Figure 8 diagnostics-13-02659-f008:**
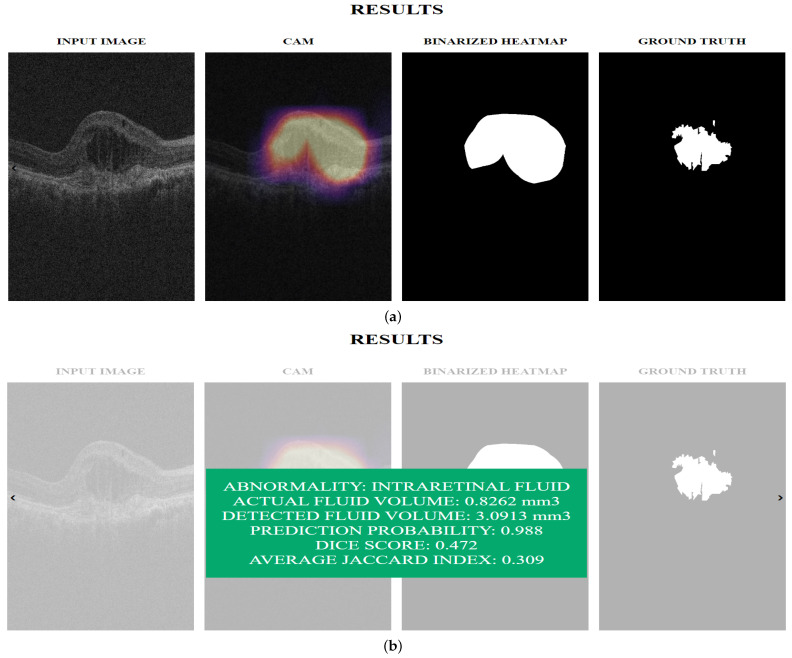
(**a**) Visualization output and (**b**) Graphical User Interface displaying the computed volume.

**Figure 9 diagnostics-13-02659-f009:**
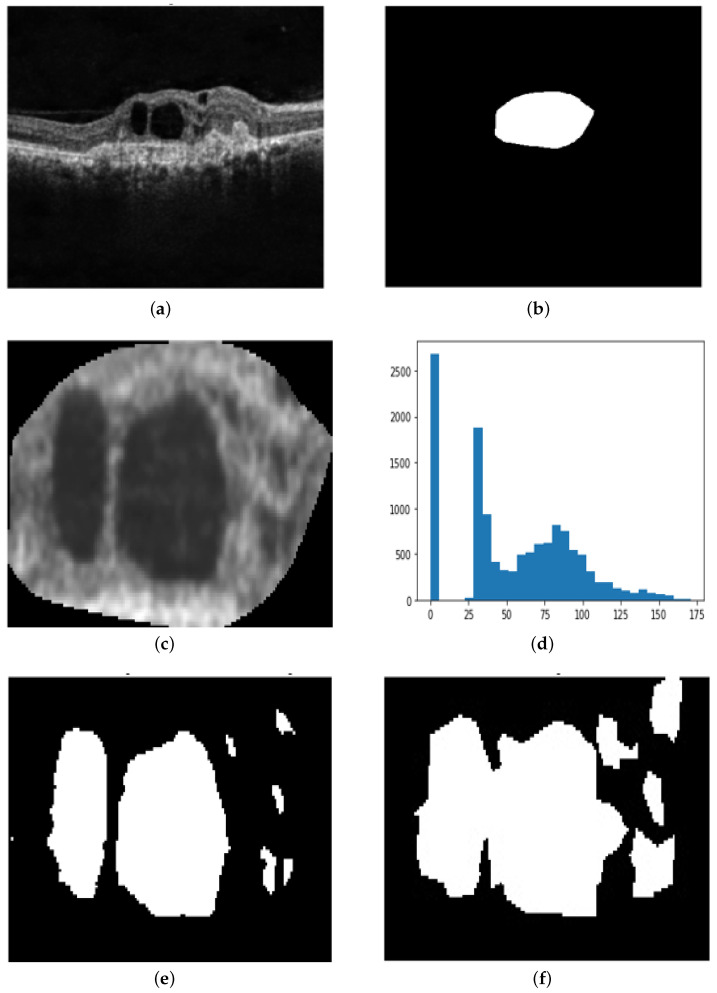
(**a**) OCT image; (**b**) binary mask obtained from visualization; (**c**) region of interest; (**d**) histogram of the region of interest; (**e**) predicted region of interest from the selective thresholding technique; and (**f**) ground-truth of the region of interest.

**Figure 10 diagnostics-13-02659-f010:**
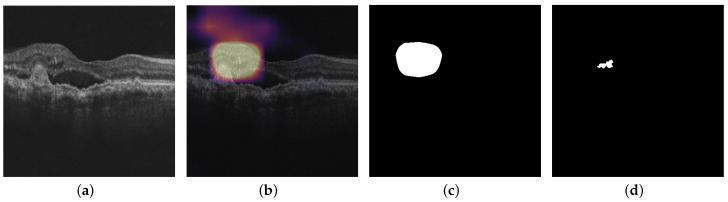
This figure shows the outputs from the Inception-ResNet-v2 GCDS model: (**a**) OCT image; (**b**) Grad-CAM heat map of OCT image for IRF; (**c**) binary map of Grad-CAM heat map; and (**d**) corresponding ground truth.

**Figure 11 diagnostics-13-02659-f011:**
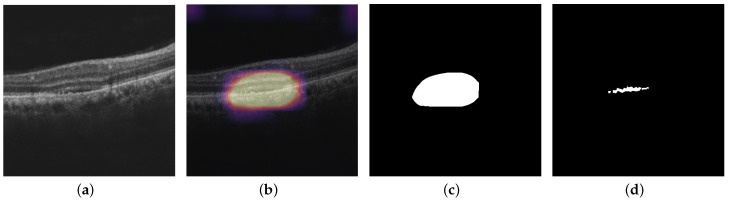
This figure shows the outputs from the Inception-ResNet-v2 GCDS model: (**a**) OCT image; (**b**) Grad-CAM heat map of OCT image for SRF; (**c**) binary map of Grad-CAM heat map; and (**d**) corresponding ground truth..

**Figure 12 diagnostics-13-02659-f012:**
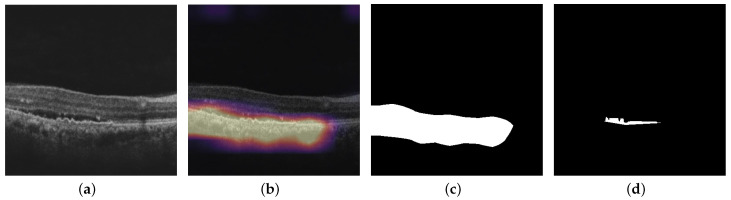
This figure shows the outputs from Inception-ResNet-v2 GCDS model: (**a**) OCT image; (**b**) Grad-CAM heat map of OCT image for PED; (**c**) binary map of Grad-CAM heat map; and (**d**) corresponding ground truth.

**Table 1 diagnostics-13-02659-t001:** Classification metrics of various models on Intraretinal Fluid pathology.

Performance Measure	Noisy Model	Inception-ResNet-v2 GCDS Model	Small Inception-ResNet-v2 GCDS Model
True Positive	8797	8407	8556
False Positive	2919	2213	2803
True Negative	40,224	40,930	40,340
False Negative	710	1100	951
Accuracy	0.9310	0.9370	0.9284
Precision	0.7508	0.7916	0.7532
Sensitivity	0.9253	0.8842	0.9000
Specificity	0.9323	0.9487	0.9350
F1 score	0.8290	0.8353	0.8201

**Table 2 diagnostics-13-02659-t002:** Classification metrics of various models on Subretinal Fluid pathology.

Performance Measure	Noisy Model	Inception-ResNet-v2 GCDS Model	Small Inception-ResNet-v2 GCDS Model
True Positive	2182	2291	2017
False Positive	613	1161	365
True Negative	49,641	49,093	49,889
False Negative	214	105	379
Accuracy	0.9842	0.9759	0.9859
Precision	0.7806	0.6636	0.8468
Sensitivity	0.9106	0.9561	0.8418
Specificity	0.9878	0.9768	0.9927
F1 score	0.8406	0.7835	0.8443

**Table 3 diagnostics-13-02659-t003:** Classification metrics of various models on Pigmented Epithelial Detachment pathology.

Performance Measure	Noisy Model	Inception-ResNet-v2 GCDS Model	Small Inception-ResNet-v2 GCDS Model
True Positive	2714	2699	2531
False Positive	2511	2795	1464
True Negative	46,434	46,150	47,481
False Negative	991	1006	1174
Accuracy	0.9334	0.9278	0.9499
Precision	0.5194	0.4913	0.6335
Sensitivity	0.7325	0.7285	0.6831
Specificity	0.9486	0.9429	0.9701
F1 score	0.6078	0.5868	0.6574

**Table 4 diagnostics-13-02659-t004:** Visualization results of various models on Intraretinal Fluid pathology.

	Noisy Model		Inception-ResNet-v2 GCDS Model		Small Inception-ResNet-v2 GCDS Model	
	**IoU**	**Dice**	**IoU**	**Dice**	**IoU**	**Dice**
Grad-CAM	0.0909	0.1556	0.1162	0.1912	0.1438	0.2325
Grad-CAM++	0.1295	0.2105	0.1498	0.2350	0.1615	0.2570
Score-CAM	0.1280	0.2083	0.1390	0.2206	0.1400	0.2274
Ablation-CAM	0.1293	0.2102	0.1497	0.2348	0.0688	0.1165
Self-MCAM	0.0664	0.1195	0.0410	0.0770	0.0345	0.0653
Ensemble-CAM	0.1278	0.2081	0.1393	0.2215	0.1365	0.2238

**Table 5 diagnostics-13-02659-t005:** Visualization results of various models on Subretinal Fluid pathology.

Performance Measure	Noisy Model		Inception-ResNet-v2 GCDS Model		Small Inception-ResNet-v2 GCDS Model	
	**IoU**	**Dice**	**IoU**	**Dice**	**IoU**	**Dice**
Grad-CAM	0.0982	0.1711	0.0811	0.1449	0.0413	0.0758
Grad-CAM++	0.0992	0.1725	0.0834	0.1485	0.0668	0.1207
Score-CAM	0.0517	0.0894	0.0875	0.1552	0.0965	0.1698
Ablation-CAM	0.0869	0.1520	0.0681	0.1218	0.0148	0.0272
Self-Matching-CAM	0.0701	0.1280	0.0758	0.1379	0.0373	0.0707
Ensemble-CAM	0.0995	0.1730	0.0859	0.1524	0.0673	0.1211

**Table 6 diagnostics-13-02659-t006:** Visualization results of various models on Pigmented Epithelial Detachment pathology.

Performance Measure	Noisy Model		Inception-ResNet-v2 GCDS Model		Small Inception-ResNet-v2 GCDS Model	
	**IoU**	**Dice**	**IoU**	**Dice**	**IoU**	**Dice**
Grad-CAM	0.1100	0.1924	0.1033	0.1819	0.0740	0.1289
Grad-CAM++	0.0174	0.0311	0.1187	0.2055	0.1028	0.1785
Score-CAM	0.0175	0.0313	0.1170	0.2033	0.1146	0.1984
Ablation-CAM	0.0176	0.0315	0.1186	0.2054	0.1039	0.1806
Self-Matching-CAM	0.0787	0.1390	0.1239	0.2119	0.1310	0.2145
Ensemble-CAM	0.0177	0.0316	0.1203	0.2082	0.1248	0.2142

**Table 7 diagnostics-13-02659-t007:** Volume computation results of various models on Intraretinal Fluid pathology for various visualization techniques using selective thresholding technique for the noisy model.

Model	Grad-CAM	Grad-CAM++	Score-CAM	Ablation-CAM	Self-Matching-CAM	Ensemble-CAM
IoU mean	0.19	0.24	0.24	0.24	0.21	0.24
IoU std	0.13	0.14	0.14	0.14	0.10	0.14
Dice mean	0.28	0.35	0.35	0.35	0.32	0.35
Dice std	0.17	0.18	0.18	0.18	0.13	0.18
Predicted vol. mean	0.62	0.35	0.31	0.35	0.27	0.35
Predicted vol. std	0.68	0.23	0.34	0.38	0.23	0.38
Ground truth vol. mean	0.15	0.15	0.15	0.15	0.15	0.15
Ground truth vol. std	0.23	0.23	0.23	0.23	0.23	0.23

**Table 8 diagnostics-13-02659-t008:** Volume computation results of various models on Intraretinal Fluid pathology for various visualization techniques using selective thresholding technique for the Inception-ResNet-v2 GCDS model.

Model	Grad-CAM	Grad-CAM++	Score-CAM	Ablation-CAM	Self-Matching-CAM	Ensemble-CAM
IoU mean	0.20	0.25	0.22	0.25	0.16	0.24
IoU std	0.17	0.20	0.20	0.20	0.08	0.20
Dice mean	0.29	0.34	0.30	0.34	0.24	0.33
Dice std	0.21	0.24	0.24	0.24	0.12	0.23
Predicted vol. mean	0.35	0.26	0.37	0.26	0.11	0.25
Predicted vol. std	0.49	0.35	0.45	0.36	0.13	0.35
Ground truth vol. mean	0.15	0.15	0.15	0.15	0.15	0.15
Ground truth vol. std	0.23	0.23	0.23	0.23	0.23	0.23

**Table 9 diagnostics-13-02659-t009:** Volume computation results of various models on Intraretinal Fluid pathology for various visualization techniques using selective thresholding technique for the small Inception-ResNet-v2 GCDS model.

Model	Grad-CAM	Grad-CAM++	Score-CAM	Ablation-CAM	Self-Matching-CAM	Ensemble-CAM
IoU mean	0.25	0.29	0.23	0.12	0.15	0.27
IoU std	0.17	0.20	0.20	0.10	0.09	0.18
Dice mean	0.34	0.40	0.31	0.17	0.23	0.37
Dice std	0.21	0.24	0.25	0.14	0.13	0.22
Predicted vol. mean	0.18	0.22	0.39	0.09	0.14	0.14
Predicted vol. std	0.18	0.21	0.34	0.09	0.16	0.14
Ground truth vol. mean	0.15	0.15	0.15	0.15	0.15	0.15
Ground truth vol. std	0.23	0.23	0.23	0.23	0.23	0.23

**Table 10 diagnostics-13-02659-t010:** Volume computation results of various models on Intraretinal Fluid pathology for various visualization techniques using region-growing technique for the noisy model.

Model	Grad-CAM	Grad-CAM++	Score-CAM	Ablation-CAM	Self-Matching-CAM	Ensemble-CAM
IoU mean	0.22	0.25	0.24	0.25	0.22	0.26
IoU std	0.16	0.17	0.17	0.17	0.13	0.17
Dice mean	0.29	0.32	0.32	0.32	0.30	0.33
Dice std	0.20	0.21	0.21	0.21	0.16	0.21
Predicted vol. mean	1.44	0.75	0.47	0.77	0.54	0.72
Predicted vol. std	1.67	0.90	0.58	0.92	0.70	0.90
Ground truth vol. mean	0.15	0.15	0.15	0.15	0.15	0.15
Ground truth vol. std	0.23	0.23	0.23	0.23	0.23	0.23

**Table 11 diagnostics-13-02659-t011:** Volume computation results of various models on Intraretinal Fluid pathology for various visualization techniques using region-growing technique for the Inception-ResNet-v2 GCDS model.

Model	Grad-CAM	Grad-CAM++	Score-CAM	Ablation-CAM	Self-Matching-CAM	Ensemble-CAM
IoU mean	0.23	0.22	0.22	0.22	0.10	0.23
IoU std	0.18	0.19	0.20	0.19	0.08	0.19
Dice mean	0.29	0.28	0.29	0.28	0.15	0.30
Dice std	0.22	0.22	0.23	0.22	0.12	0.22
Predicted vol. mean	0.82	0.76	0.85	0.77	0.78	0.66
Predicted vol. std	1.08	0.84	0.99	0.87	0.79	0.85
Ground truth vol. mean	0.15	0.15	0.15	0.15	0.15	0.15
Ground truth vol. std	0.23	0.23	0.23	0.23	0.23	0.23

**Table 12 diagnostics-13-02659-t012:** Volume computation results of various models on Intraretinal Fluid pathology for various visualization techniques using region-growing technique for the small Inception-ResNet-v2 GCDS model.

Model	Grad-CAM	Grad-CAM++	Score-CAM	Ablation-CAM	Self-Matching-CAM	Ensemble-CAM
IoU mean	0.23	0.23	0.15	0.14	0.15	0.24
IoU std	0.18	0.19	0.15	0.14	0.10	0.19
Dice mean	0.30	0.30	0.20	0.18	0.19	0.31
Dice std	0.23	0.24	0.20	0.18	0.15	0.23
Predicted vol. mean	0.87	0.62	0.94	0.94	0.65	0.60
Predicted vol. std	1.16	0.66	0.85	0.97	0.85	0.87
Ground truth vol. mean	0.15	0.15	0.15	0.15	0.15	0.15
Ground truth vol. std	0.23	0.23	0.23	0.23	0.23	0.23

**Table 13 diagnostics-13-02659-t013:** Volume computation results of various models on Subretinal Fluid pathology for various visualization techniques using selective thresholding technique for the noisy model.

Model	Grad-CAM	Grad-CAM++	Score-CAM	Ablation-CAM	Self-Matching-CAM	Ensemble-CAM
IoU mean	0.16	0.17	0.06	0.15	0.16	0.17
IoU std	0.11	0.11	0.10	0.10	0.09	0.11
Dice mean	0.25	0.26	0.09	0.24	0.26	0.26
Dice std	0.15	0.15	0.14	0.13	0.12	0.15
Predicted vol. mean	0.66	0.62	5.98	0.55	0.36	0.61
Predicted vol. std	0.96	0.87	6.81	0.72	0.52	0.87
Ground truth vol. mean	0.37	0.37	0.37	0.37	0.37	0.37
Ground truth vol. std	0.71	0.71	0.71	0.71	0.71	0.71

**Table 14 diagnostics-13-02659-t014:** Volume computation results of various models on Subretinal Fluid pathology for various visualization techniques using selective thresholding technique for the Inception-ResNet-v2 GCDS model.

Model	Grad-CAM	Grad-CAM++	Score-CAM	Ablation-CAM	Self-Matching-CAM	Ensemble-CAM
IoU mean	0.09	0.11	0.13	0.09	0.13	0.11
IoU std	0.09	0.11	0.12	0.08	0.10	0.12
Dice mean	0.15	0.16	0.19	0.14	0.21	0.17
Dice std	0.13	0.15	0.17	0.11	0.14	0.16
Predicted vol. mean	0.63	0.56	0.46	0.48	0.12	0.52
Predicted vol. std	0.85	0.71	0.61	0.57	0.21	0.67
Ground truth vol. mean	0.37	0.37	0.37	0.37	0.37	0.37
Ground truth vol. std	0.71	0.71	0.71	0.71	0.71	0.71

**Table 15 diagnostics-13-02659-t015:** Volume computation results of various models on Subretinal Fluid pathology for various visualization techniques using selective thresholding technique for the small Inception-ResNet-v2 GCDS model.

Model	Grad-CAM	Grad-CAM++	Score-CAM	Ablation-CAM	Self-Matching-CAM	Ensemble-CAM
IoU mean	0.07	0.13	0.18	0.01	0.14	0.10
IoU std	0.07	0.12	0.14	0.02	0.09	0.11
Dice mean	0.11	0.21	0.27	0.02	0.21	0.15
Dice std	0.11	0.16	0.19	0.03	0.15	0.15
Predicted vol. mean	0.22	0.17	0.19	0.13	0.19	0.69
Predicted vol. std	0.36	0.30	0.33	0.18	0.25	0.66
Ground truth vol. mean	0.37	0.37	0.37	0.37	0.37	0.37
Ground truth vol. std	0.71	0.71	0.71	0.71	0.71	0.71

**Table 16 diagnostics-13-02659-t016:** Volume computation results of various models on Subretinal Fluid pathology for various visualization techniques using region-growing technique for the noisy model.

Model	Grad-CAM	Grad-CAM++	Score-CAM	Ablation-CAM	Self-Matching-CAM	Ensemble-CAM
IoU mean	0.31	0.30	0.12	0.28	0.28	0.31
IoU std	0.19	0.18	0.16	0.17	0.16	0.19
Dice mean	0.39	0.38	0.14	0.36	0.36	0.39
Dice std	0.22	0.21	0.19	0.20	0.19	0.22
Predicted vol. mean	1.16	1.10	9.08	1.01	1.14	1.08
Predicted vol. std	1.16	1.15	8.78	1.40	1.48	1.54
Ground truth vol. mean	0.37	0.37	0.37	0.37	0.37	0.37
Ground truth vol. std	0.71	0.71	0.71	0.71	0.71	0.71

**Table 17 diagnostics-13-02659-t017:** Volume computation results of various models on Subretinal Fluid pathology for various visualization techniques using region-growing technique for the Inception-ResNet-v2 GCDS model.

Model	Grad-CAM	Grad-CAM++	Score-CAM	Ablation-CAM	Self-Matching-CAM	Ensemble-CAM
IoU mean	0.12	0.17	0.22	0.14	0.20	0.17
IoU std	0.13	0.15	0.16	0.13	0.15	0.14
Dice mean	0.16	0.22	0.29	0.19	0.26	0.22
Dice std	0.16	0.18	0.19	0.16	0.19	0.18
Predicted vol. mean	0.84	0.88	1.16	1.58	0.89	4.39
Predicted vol. std	1.05	0.87	1.05	1.16	1.17	4.21
Ground truth vol. mean	0.37	0.37	0.37	0.37	0.37	0.37
Ground truth vol. std	0.71	0.71	0.71	0.71	0.71	0.71

**Table 18 diagnostics-13-02659-t018:** Volume computation results of various models on Subretinal Fluid pathology for various visualization techniques using region-growing technique for the small Inception-ResNet-v2 GCDS model.

Model	Grad-CAM	Grad-CAM++	Score-CAM	Ablation-CAM	Self-Matching-CAM	Ensemble-CAM
IoU mean	0.13	0.19	0.27	0.13	0.21	0.14
IoU std	0.18	0.18	0.19	0.11	0.17	0.16
Dice mean	0.17	0.25	0.35	0.18	0.27	0.19
Dice std	0.22	0.22	0.23	0.15	0.20	0.20
Predicted vol. mean	0.93	0.66	0.51	1.75	0.96	0.62
Predicted vol. std	1.51	1.02	0.78	1.20	1.55	0.86
Ground truth vol. mean	0.37	0.37	0.37	0.37	0.37	0.37
Ground truth vol. std	0.71	0.71	0.71	0.71	0.71	0.71

**Table 19 diagnostics-13-02659-t019:** Volume computation results of various models on Pigmented Epithelial Detachment pathology for various visualization techniques using selective thresholding technique for the noisy model.

Model	Grad-CAM	Grad-CAM++	Score-CAM	Ablation-CAM	Self-Matching-CAM	Ensemble-CAM
IoU mean	0.02	0.00	0.00	0.00	0.01	0.00
IoU std	0.03	0.01	0.01	0.01	0.02	0.01
Dice mean	0.04	0.01	0.01	0.01	0.02	0.01
Dice std	0.05	0.02	0.02	0.02	0.03	0.02
Predicted vol. mean	1.52	0.61	0.61	0.61	3.67	0.61
Predicted vol. std	1.28	0.67	0.67	0.67	3.31	0.67
Ground truth vol. mean	0.62	0.62	0.62	0.62	0.62	0.62
Ground truth vol. std	0.48	0.48	0.48	0.48	0.48	0.48

**Table 20 diagnostics-13-02659-t020:** Volume computation results of various models on Pigmented Epithelial Detachment pathology for various visualization techniques using selective thresholding technique for the Inception-ResNet-v2 GCDS model.

Model	Grad-CAM	Grad-CAM++	Score-CAM	Ablation-CAM	Self-Matching-CAM	Ensemble-CAM
IoU mean	0.01	0.02	0.02	0.02	0.01	0.02
IoU std	0.02	0.02	0.02	0.02	0.01	0.02
Dice mean	0.03	0.03	0.03	0.03	0.01	0.03
Dice std	0.04	0.04	0.04	0.04	0.02	0.04
Predicted vol. mean	0.84	0.52	0.55	0.52	0.20	0.49
Predicted vol. std	0.85	0.53	0.56	0.53	0.28	0.54
Ground truth vol. mean	0.62	0.62	0.62	0.62	0.62	0.62
Ground truth vol. std	0.48	0.48	0.48	0.48	0.48	0.48

**Table 21 diagnostics-13-02659-t021:** Volume computation results of various models on Pigmented Epithelial Detachment pathology for various visualization techniques using selective thresholding technique for the small Inception-ResNet-v2 GCDS model.

Model	Grad-CAM	Grad-CAM++	Score-CAM	Ablation-CAM	Self-Matching-CAM	Ensemble-CAM
IoU mean	0.02	0.02	0.01	0.01	0.01	0.01
IoU std	0.03	0.02	0.02	0.02	0.01	0.02
Dice mean	0.03	0.03	0.02	0.02	0.02	0.02
Dice std	0.05	0.04	0.04	0.04	0.02	0.03
Predicted vol. mean	0.99	0.76	0.43	0.35	0.19	0.28
Predicted vol. std	0.90	0.70	0.40	0.33	0.25	0.28
Ground truth vol. mean	0.62	0.62	0.62	0.62	0.62	0.62
Ground truth vol. std	0.48	0.48	0.48	0.48	0.48	0.48

**Table 22 diagnostics-13-02659-t022:** Volume computation results of various models on Pigmented Epithelial Detachment pathology for various visualization techniques using region-growing technique for the noisy model. GT stands for ground truth provided by an expert.

Model	Grad-CAM	Grad-CAM++	Score-CAM	Ablation-CAM	Self-Matching-CAM	Ensemble-CAM
IoU mean	0.04	0.04	0.01	0.01	0.04	0.05
IoU std	0.03	0.04	0.01	0.02	0.05	0.06
Dice mean	0.06	0.07	0.01	0.01	0.07	0.08
Dice std	0.05	0.06	0.02	0.02	0.07	0.05
Predicted vol. mean	4.99	4.85	2.26	12.31	5.85	2.36
Predicted vol. std	4.05	4.46	1.46	8.34	4.57	1.99
GT vol. mean	0.62	0.62	0.62	0.62	0.62	0.62
GT vol. std	0.48	0.48	0.48	0.48	0.48	0.48

**Table 23 diagnostics-13-02659-t023:** Volume computation results of various models on Pigmented Epithelial Detachment pathology for various visualization techniques using region-growing technique for the Inception-ResNet-v2 GCDS model.

Model	Grad-CAM	Grad-CAM++	Score-CAM	Ablation-CAM	Self-Matching-CAM	Ensemble-CAM
IoU mean	0.06	0.04	0.03	0.05	0.05	0.05
IoU std	0.04	0.04	0.03	0.04	0.04	0.04
Dice mean	0.09	0.07	0.05	0.09	0.09	0.09
Dice std	0.06	0.06	0.04	0.06	0.06	0.06
Predicted vol. mean	2.24	4.85	1.52	2.02	3.10	1.76
Predicted vol. std	1.89	4.46	1.56	1.07	3.22	1.49
Ground truth vol. mean	0.62	0.62	0.62	0.62	0.62	0.62
Ground truth vol. std	0.48	0.48	0.48	0.48	0.48	0.48

## Data Availability

The data were obtained from a Sankara Nethralaya eye hospital, and the patient data will be maintained with confidentiality. The data will be shared upon request and can be provided only after approval from the institutional review board.

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
