# Peer review of "Deep-Learning-Based Visualization and Volumetric Analysis of Fluid Regions in Optical Coherence Tomography Scans"

_diagnostics, 2023, doi:10.3390/diagnostics13162659_

Round 1
Reviewer 1 Report
The manuscript by Harishwar Reddy K et al. demonstrated the results of the classification of three pathologies for retinal OCT images using different neural network models. Besides, the authors employed visualization techniques to represent the feature significance from trained model parameters. The structure of the manuscript is well organized. However, the reviewer does have a few comments that would like the authors to address or elaborate on, as listed below.
1. In the description of lines 119~120 and Figure 4, is the noisy model trained by the inception ResNet v2? In Figure 4, there is a step for preprocessing the OCT images which seems not to be depicted in section 2.5. Was this step just involved in resizing the denoised or noisy OCT images?
2. When training these models, did the authors use k-fold cross-validation? Approximately how much time does it take to train these models?
3. Figures 11 to 14 show the results for visualization of three different retinal pathologies, including IRF, SRF, and PED, respectively. However, for the OCT images of healthy retinas, does the region of the binary maps of the Grad-CAM heatmap correspond to a similar or identical location in Figures 11(b) to 14(b), respectively?
4. Please check Figures 11 to 14, and should the descriptions of images (c) and (d) be swapped? (c) seems to be the binary map of the Grad-CAM heatmap, while (d) seems to be the corresponding ground truth.
5. For tables 7 to 24, the value of IoU and dice are far from 1, and the predicted volume is larger than the counterpart of ground truth. Do the authors have any explanation for that, and what factors contribute to such outcomes? Is it possible to further improve and make the predicted volume results closer to the ground truth?
Reviewer 2 Report
The paper focuses on deep learning-based visualization techniques to compute the retinal fluid in OCT scans of patients affected by Intraretinal Fluid (IRF), Subretinal Fluid (SRF), and Pigmented Epithelial Detachment (PED).
General remarks
The proposed study is interesting and challenging, however, there are some issues that should be properly fixed. First of all, the state of the art should be investigated more in-depth and the number of references should be increased. Moreover, throughout the paper, the reader can find an improper choice of words (e.g., mild, seed), as well as unexplained acronyms (e.g., AI, CAM, C-scan).
The specific remarks are listed below.
Specific remarks
Section 1 – INTRODUCTION
· The state of the art should be investigated more in-depth to provide a comprehensive overview of the current scientific literature regarding the proposed topic. In particular, the limitations and disadvantages of the solutions already present in the literature should be outlined to highlight the novelty of this work.
· “Intraretinal Fluid (IRF), Subretinal Fluid (SRF), and Pigmented Epithelial Detachment (PED) are three of the most prevalent retinal pathologies”: can you provide references?
· “In this paper, we model the speckle noise as gamma distribution …”: why did you choose this distribution? What are the advantages of the gamma distribution in this application, and which are the limitations of the others used in the literature?
· “Deep learning has produced remarkable advancements in the field of healthcare, …”: can you provide proper references?
· “Deep learning-based classifiers are used to detect the relationship between the multiple input features in the image and arrive at a consensus” and “Explainability in AI has now become an indispensable part of the deployment pipeline”: these sentences are not clear.
· “To the best of our knowledge, ours is the first attempt at adopting the visualization outcome of a deep learning model for retinal fluid volume computation instead of a segmentation output”: what can be found in the current literature for retinal fluid volume computation? This issue is interesting and deserves a detailed description.
· Figure 1: I suggest modifying it to better distinguish the three pathologies in the OCT image.
Section 2 – MATERIALS AND METHODS
· subsection 2.1.2: please, add the mathematical expression for the computation of both the Jaccard Index/Intersection over Union (IoU) and Dice score, as well as references.
· subsection 2.2: please, provide details about Inception-ResNet-v2.
· Figure 2: the block diagrams should be described better in the text.
· subsection 2.4: what did you mean by mild denoising?
· “After experimenting with several noise models, we have found the gamma distribution to be an appropriate model for the noise in OCT images”: how has been evaluated the appropriate model?
· “We carry out denoising using Gated Convolution and Deconvolution Structure (GCDS) model”: what are the advantages of such a model?
· Figure 3: the images are not mentioned in the text.
· “… we noisy OCT images are denoised and used for training the classification model”: the sentence is not clear.
· “We used a learning rate of 10-4, and Adam as the optimization method”: this stage of the study is interesting and deserves a detailed description.
· Figure 6: the images are not mentioned in the text the caption is incomplete.
· “If the prediction score is more than 0.5, we pass …”: how did you choose a 0.5 value?
· Figure 8b: it is difficult to read the text in the window.
· subsection 2.7.1: the standard region-growing algorithm deserves a detailed description and proper references.
· Equation 2: the quantities in the equation should be explained.
· subsection 2.7.2: why did you choose an intensity value of 50? A standardized procedure to compute the intensity range should be preferred, otherwise, how can you adapt the threshold values for the processing of OCT images acquired through different systems?
Section 3 – RESULTS
· subsection 3.1: what did you mean by abnormal and normal prediction?
· Equation 8: what did you mean by recall? This index has not been introduced.
· Tables 1-3: I suggest providing TP, FP, TN and FN in percentage. Moreover, the tables are not mentioned in the section.
· Figures 10-12: the images are not mentioned in the text.
· Tables 7-24: the number of significant digits used to express the results is wrong. Moreover, the tables are not mentioned in the section.
· Results obtained from the processing of the images acquired through the PRIMUS machine are completely missing.
Section 4 – DISCUSSION
· Results are not sufficiently discussed. This section should be rewritten and properly extended.
· Lines 266-277: these considerations should be included in the introduction section.
Throughout the paper, the reader can find an improper choice of words (e.g., mild, seed), as well as unexplained acronyms (e.g., AI, CAM, C-scan).
Round 2
Reviewer 1 Report
The author's response already answers the reviewer's question.
Author Response
The reviewers has no comments to answer
Reviewer 2 Report
It was very difficult to detect and follow the required text changes because they have not been highlighted in the revised version of the paper in any way. Moreover, the authors did not respond to all my comments. The issues that must be addressed yet and included in the paper are listed in the following.
Section 1 – INTRODUCTION
· The state of the art should be investigated more in-depth to provide a comprehensive overview of the current scientific literature regarding the proposed topic. In particular, the limitations and disadvantages of the solutions already present in the literature should be outlined to highlight the novelty of this work.
· “Deep learning-based classifiers are used to detect the relationship between the multiple input features in the image and arrive at a consensus” and “Explainability in AI has now become an indispensable part of the deployment pipeline”: these sentences are still not clear. Please, rewrite them.
· “To the best of our knowledge, ours is the first attempt at adopting the visualization outcome of a deep learning model for retinal fluid volume computation instead of a segmentation output”: what can be found in the current literature for retinal fluid volume computation? This issue is interesting and deserves a detailed description.
· Figure 1: I suggest modifying it to better distinguish the three pathologies in the OCT image.
Section 2 – MATERIALS AND METHODS
· “We carry out denoising using Gated Convolution and Deconvolution Structure (GCDS) model”: what are the advantages of such a model? Please, specify them in the text.
· “We used a learning rate of 10-4, and Adam as the optimization method”: this stage of the study is interesting and deserves a detailed description in the paper.
· Equation 2: the quantities and the coefficient in the equation should be explained one by one.
Section 3 – RESULTS
· Tables 7-24: the number of significant digits used to express the results is still wrong. The significant figures depend on the measurement uncertainty that should be expressed with one significant digit unless its value is 1 or 2 (because in that case rounding off to one digit may lead to a significant loss of information). I suggest you the following references:
[1] Taylor J.R. "An Introduction to Error Analysis: The Study of Uncertainties in Physical Measurements", 2nd edition, University Science Books, 1997.
[2] GUM: Guide to the Expression of Uncertainty in Measurement (https://www.bipm.org/en/publications/guides/gum.html)
· If you were unable to conduct experiments on PRIMUS images, any reference to the PRIMUS machine must be removed from the paper and possibly included in future developments of the proposed study.
Section 4 – DISCUSSION
· Results are not sufficiently discussed. This section should be properly extended. The standard deviation values are very often higher than the corresponding mean values. What could this be due to? How can be the results interpreted when this issue occurs? Please, discuss it in the text.
Moderate editing of English language required.
Round 3
Reviewer 2 Report
The authors replied to all items requested.
Moderate editing of English language required.